# Comparative Pathogenesis, Genomics and Phylogeography of Mousepox

**DOI:** 10.3390/v13061146

**Published:** 2021-06-15

**Authors:** Carla Mavian, Alberto López-Bueno, Rocío Martín, Andreas Nitsche, Antonio Alcamí

**Affiliations:** 1Centro de Biología Molecular Severo Ochoa, Consejo Superior de Investigaciones Científicas, Campus de Cantoblanco, Universidad Autónoma de Madrid, Nicolás Cabrera 1, 28049 Madrid, Spain; cmavian@ufl.edu (C.M.); alopezbueno@cbm.csic.es (A.L.-B.); mrmartin@cbm.csic.es (R.M.); 2Centre for Biological Threats and Special Pathogens, Highly Pathogenic Viruses (ZBS1), Robert Koch Institute, 13353 Berlin, Germany; NitscheA@rki.de

**Keywords:** mousepox, poxvirus pathogenesis, genome sequence, virulence, A-type inclusion bodies

## Abstract

Ectromelia virus (ECTV), the causative agent of mousepox, has threatened laboratory mouse colonies worldwide for almost a century. Mousepox has been valuable for the understanding of poxvirus pathogenesis and immune evasion. Here, we have monitored in parallel the pathogenesis of nine ECTVs in BALB/cJ mice and report the full-length genome sequence of eight novel ECTV isolates or strains, including the first ECTV isolated from a field mouse, ECTV-MouKre. This approach allowed us to identify several genes, absent in strains attenuated through serial passages in culture, that may play a role in virulence and a set of putative genes that may be involved in enhancing viral growth in vitro. We identified a putative strong inhibitor of the host inflammatory response in ECTV-MouKre, an isolate that did not cause local foot swelling and developed a moderate virulence. Most of the ECTVs, except ECTV-Hampstead, encode a truncated version of the P4c protein that impairs the recruitment of virions into the A-type inclusion bodies, and our data suggest that P4c may play a role in viral dissemination and transmission. This is the first comprehensive report that sheds light into the phylogenetic and geographic relationship of the worldwide outbreak dynamics for the ECTV species.

## 1. Introduction

Orthopoxviruses have played an important role in human history: variola virus (VARV), the causative agent of smallpox, was one of the most lethal human pathogens; vaccinia virus (VACV) and cowpox virus (CPXV) were key for the discovery of vaccination [1,2,3,4]. Today, 40 years after the eradication of smallpox, VARV still represents a threat as a bioterrorist weapon, and CPXV and monkeypox virus are the main cause of zoonotic poxvirus infections [5,6,7]. The orthopoxvirus ectromelia virus (ECTV) is the causative agent of mousepox, an acute exanthematous disease of mice that resembles smallpox and was used as a model for studying orthopoxvirus infections [8,9,10]. Mousepox was a serious threat to laboratory mouse colonies in the past [11,12,13,14], but outbreaks of mousepox in laboratory mouse colonies have been eliminated thanks to strict surveillance and improvements in animal house facilities with pathogen contention racks, filters and disinfections [11]. The first ECTV outbreak was reported in 1930 in the National Institute for Medical Research in Hampstead (London, UK) and was caused by the ECTV-Hampstead (ECTV-H) isolate (Table 1). The disease was named “infectious ectromelia” because of the characteristic foot amputation observed in mice that recovered from infection [8]. Sixty passages in chorioalantoic membranes (CAMs) of ECTV-H resulted in the attenuated strain ECTV-Hampstead Egg (ECTV-HE) that shows a considerable reduction in virulence in outbred mice [15], and the same origin was proposed for the attenuated strain ECTV-Mill Hill (ECTV-MH) [16]. After this first case, numerous outbreaks were reported in animal facilities across Europe and in 1946 the highly virulent ECTV-Moscow (ECTV-M) was isolated from an outbreak in Moscow [17]. ECTV-H and ECTV-M were used by F. Fenner in early studies to characterize mousepox pathogenesis [15,17,18,19,20,21,22,23]. Since then, numerous ECTV outbreaks were registered in Europe: in 1976 there were outbreaks in mouse colonies in Munich and Nuremberg, Germany, where ECTV-MP1 (ECTV-M1) and ECTV-MP4 (ECTV-M4) were isolated, respectively, and later in 1994 the isolate ECTV-MP5 (ECTV-M5) was responsible from an outbreak in Wien, Austria [24,25]. In 1986 and 1988 two outbreaks in Warsaw, Poland, were firstly described as ECTV isolates, but their genome sequencing demonstrated that they corresponded to VACV [26,27]. In the 1950s, mousepox outbreaks were also described in China and Japan, where another laboratory outbreak was registered in Ishibashi and the causative agent was the isolate ECTV-Ishibashi (ECTV-I), which was attenuated by more than 30 passages in cell culture [28]. A study of sera from 150 wild mice around Melbourne did not detect exposure to ECTV, suggesting the absence of this virus in the Australian continent [29]. In North America, ECTV was not enzootic, and the first official ECTV outbreak was described in Yale University in 1953 [30]. Occasional outbreaks were registered until 1979–1980 when the ECTV-NIH79 isolate caused a series of devastating outbreaks that led the National Institute of Health to create a surveillance committee that prohibited the introduction of ECTV in USA [31,32]. The last outbreaks in this continent were reported in 1995 and 1999 [33,34]. The isolate ECTV-Naval (ECTV-N), responsible for the outbreak of 1995, was isolated at the US Naval Medical Research Institute in Bethesda (Maryland, USA) and derived from a commercial serum imported from China that caused a lethal disease in BALB/cJ mice [34,35]. The ECTV outbreak of 1999 at the Weill Medical College of Cornell University in New York (USA) was caused by ECTV-Cornell, which was found to be genetically identical to ECTV-N [34,36]. The source of this virus was also linked to commercial mouse serum imported from China, consistent with the genomes of ECTV-N and ECTV-C sharing more than 99% of nucleotide (nt) sequence identity with erythomelalgia-related poxvirus (ERPV), an ECTV isolated in China [34,37].

The mouse is the known natural host of ECTV, aside from one report in the silver fox [38], and it has been proposed that populations of wild mice in Europe can represent the reservoir of ECTV as the virus can transmit from wild mice to laboratory-mouse colonies [12,39]. However, to our knowledge, there is only one report describing the isolation of ECTV from a wild mouse, but this identification was not further confirmed by genetic analyses [39]. Phylogenetic analysis of the highly conserved central region of poxvirus genomes identified ECTV in a unique branch of the orthopoxvirus as the most distant species within this genus [40]. The genome of ECTV encodes a highly specialized array of proteins that interact with the mouse immune system and play a critical role in viral pathogenesis and host defense [34,37,41]. The availability of a mouse model for ECTV has facilitated the in vivo characterization of many of these genes by following a knocking-down strategy [42,43,44,45,46]. The restricted number of ECTV isolates with full-length genome sequences available, currently only ECTV-M, ECTV-N, and ERPV [34,37,41], has limited the identification of genes involved in virulence by correlating gene-content and virulence from different virus isolates, a strategy successfully followed with other poxviruses [47,48,49]. Despite the relevance of ECTV outbreaks in laboratory mouse colonies and the importance of mousepox in classical studies of viral pathogenesis, there are no studies addressing the genomic diversity of ECTV isolates.

In this study, we present a comparative analysis of the pathogenesis in BALB/cJ mice and the full-length genome sequence of a collection of ECTV isolates from worldwide laboratory outbreaks and ECTV strains attenuated after serial passage in cell culture or CAMs. We also report the first characterization of an ECTV isolated from a wild mouse in 2008 in the region of Krefeld (Germany), named ECTV-MouKre (ECTV-MK). Comparison of the tissue culture growth and virulence in susceptible mice of the ECTV isolates, linked to their genomic sequence, has identified putative virulence determinants and new immunomodulatory molecules encoded by ECTV. Differences in the ability of the ECTV isolates to recruit virions to A-type inclusion (ATI) bodies has allowed us to address the role of ATI bodies in ECTV transmission. Finally, by calibrating a molecular clock and applying Bayesian ancestral reconstruction, this study provides phylogenetic support to trace connections between the historical outbreaks of ECTV in laboratory mouse colonies.

## 2. Materials and Methods

### 2.1. Cells and Viruses

Monkey kidney BS-C-1 cells (ATCC: CCL-26) were used for virus amplification and preparation of semi-purified viral stocks as described [34]. ECTV-H [8] and ECTV-MH [16] (original stocks from K. Dumbell) were supplied by J. Williamson (St. Mary’s Hospital, Imperial College School of Medicine, London, United Kingdom); ECTV-HE was provided by A. Mullbacher (John Curtin School of Medical Research, Australian National University, Canberra, Australia) [15]; ECTV-I was provided by Y. Ichihashi (Faculty of Medicine, Niigata University, Niigata, Japan) [28]; ECTV-M1 [24], ECTV-M4 and ECTV-M5 [25] were supplied by H. Meyer (Institute of Microbiology, Federal Armed Forces Medical Academy, Munich, Germany); ECTV-MK was isolated in 2008 from wild mice in the region of Krefeld, Germany, by A. Nitsche (The Robert Koch Institute, Berlin, Germany); ECTV-M (ECTV-Moscow-3-P2) is a plaque-purified isolate provided by R. M. L. Buller (School of Medicine, Saint Louis University) [17,41]; ECTV-N is a plaque-purified isolate [34] derived from the original stock of the Naval Medical Research Institute outbreak (Bethesda, MD, USA) provided by R. M. L. Buller [35]. A list of the viruses used in the study is given in Table 1.

Virus titers were determined by plaque assays on BS-C-1 cell monolayers grown. Briefly, cells were infected for 2 h with different virus doses in Dulbecco’s Modified Eagle Medium DMEM 2.5% heat-inactivated fetal bovine serum, and after 2 h the medium was replaced by a semi-solid medium consisting of carboxymethylcellulose (1500–3000 centipoise; Sigma-Aldrich, Saint Louis, MO, USA) at 0.15% weight/volume (*w*/*v*) and carboxymethylcellulose (50–200 centipoise; Sigma-Aldrich, Saint Louis, MO, USA) at 0.15% (*w*/*v)* in Dulbecco’s Modified Eagle Medium DMEM 2.5% heat-inactivates fetal bovine serum. Cell monolayers were stained 5 days later with 2% crystal violet and 2% formaldehyde to identify virus plaques.

One-step growth curve experiments were assessed on BS-C-1 cells infected at a multiplicity of infection of 3.5–10 plaque-forming unit (pfu) per cell (pfu/cell) and viral titers were determined at 0, 3, 7, 11 and 24 h post-infection (hpi) as described [34].

### 2.2. In Vivo Experiments

Groups of 6–7 weeks old female BALB/cJ or DBA/2 mice (Charles River Laboratories) were anesthetized using isofluorane and infected subcutaneously (s. c.) in the left hind footpad with different doses of ECTV or phosphate buffer saline (PBS) as control. Signs of illness and body weight were monitored daily for 19 days post-infection (dpi). Mice were housed in biosafety level 3 ventilated racks (Allentown) with free access to food and water and 12 h light/12 h dark cycle.

To assess viral dissemination, mice were infected with 10 pfu of ECTV-M or ECTV-H and sacrificed to obtain popliteal lymph nodes, spleen and liver at 3 dpi, and spleen, liver and lung at 5 and 7 dpi. Transmission of the virus was analyzed in groups of mice infected with 10^3^ pfu of ECTV-M or ECTV-H. Two of these infected mice were transferred to a cage containing four uninfected mice (exposed group) at 2, 4 and 7 dpi to assess early (group 1), intermediate (group 2) or late (group 3) transmission, respectively. Three days post-exposure (dpe), the infected mice were removed from the cage. Exposed mice were sacrificed 7 dpe, and spleen and liver viral load were determined by plaque assay as indicator of viral transmission. Organs were PBS-washed and homogenized in PBS to 10% (*w*/*v*) with a 5 mm steel sphere using the TissueLyser II (Qiagen) with 1 min at 30 Hz pulses twice for the liver, four times for the lung and five times for the spleen. Statistical significance was inferred applying a non-parametric two tail Mann-Whitney test using GraphPad Prism 9 (GraphPad Software, San Diego, CA, USA).

### 2.3. Electron Microscopy

BS-C-1 cells infected at a multiplicity of infection of 10 pfu/cell or spleen samples from infected BALB/cJ mice were fixed with 4% paraformaldehyde and 2% glutaraldehyde in 1× PBS for 60 min at room temperature. Postfixation was carried out with 1% OsO_4_ and 1% K_3_Fe(CN)_6_ in distilled H_2_O at 4 °C for 60 min, and subsequently fixed again with 0.15% tannic acid in PBS. Fixed samples were incubated with 2% uranyl acetate for 1 h, dehydrated with absolute ethanol, flat-embedded in Epon resin (Epoxi, TAAB812, TAAB Laboratories, Berkshire, England) overnight, replaced once and finally polymerized at 60 °C for 48 h. Images were captured under a JEM1010 electronic microscope coupled to a TemCam-F416 (4 K × 4 K) camera (TVIPS, Gauting, Germany) under a magnification of 10,000×. Only cells that were clearly infected, as harboring virions in the cell, were quantified. One hundred cells per ECTV isolate from different sections were captured and analyzed to quantify the number of cells presenting an ATI bodies positive phenotype.

### 2.4. Genome Sequencing and Assembly

Two µg of viral DNA, obtained from semi-purified viral stocks as described [34], were used for the construction of a library using the GS-FLX Titanium system (454 Life Sciences, Roche, Branford, CT, USA) and sequenced with a FLX Genome Sequencer in the Scientific Park of Madrid, Spain. Reads were *de novo* assembled and also mapped to ECTV-M or ECTV-N reference genomes with Newbler 2.5.3 (Roche Diagnostics, Branford, CT, USA). For Illumina sequencing, five µg of viral DNA of ECTV-M was used to construct a TruSeq library and reads obtained with a Genome Analyzer IIx in the Scientific Park of Madrid, Spain. These Illumina reads were mapped to ECTV-M genome (AF012825.2) as reference using Bowtie2 with default parameters (http://bowtie-bio.sourceforge.net/bowtie2/manual.shtml; Version 2.1.0) (Table 2). A PCR amplification of the region containing the Direct Repeat III (DRIII) region of ECTV-MK was carried out as described [27]. The product was analyzed in a 2% agarose gel with a 100 bp ladder marker (Invitrogen) considering the most intense band for the estimation of the number of repeats.

### 2.5. Genome Annotation, Analysis and Comparison

The genomes were annotated using Artemis (Wellcome Trust Sanger Institute, United Kingdom) as described [34,50]. Genes and pseudogenes were named with a three-letter acronym for each ECTV strain (the first two letters stand for ECTV and the third for the isolate/strain) and a consecutive number from the left to the right region of the genome. Pseudogenes are indicated with a P after the number. Genome identity percentage and nucleotide differences among genomes were obtained by full-length genome alignment using Geneious Pro 6.1 (Biomatters, Auckland, New Zealand). The graphical representation of the genomic differences (smooth plot) among the genomes of the new ECTV isolates/strains, and the published genomes of ECTV-N (KJ563295), ERPV (JQ410350.1) and CPXV-BR (AF482758.2), was obtained by BLASTz alignment using the mVista software (http://genome.lbl.gov/vista/index.shtml, University of California, Berkeley, California, USA) and ECTV-M (NC_004105.1) as reference. Dot plot analysis of ECTV-H and ECTV-HE genomes was performed using the JDotter program (http://athena.bioc.uvic.ca/virology-ca-tools/jdotter/, University of Victoria, Victoria, Canada). The proteins encoded by each gene from all the ECTV strains were individually aligned with Clustaw2 (http://www.ebi.ac.uk/Tools/msa/clustalw2/, EMBL-EBI in Hinxton, Cambridge, UK).

### 2.6. Bayesian Phylogeography Analysis

The presence of temporal signal (R^2^ = 0.74, correlation coefficient = 0.86) was assessed using TempEst v1.7 (University of Edinburgh, Edinburgh, UK) [51] (Appendix A) and a maximum likelihood tree with best-fit HKY model chosen according to Bayesian Information Criteria (BIC) implemented in the IQ-TREE v1.6.12 (University of Vienna, Vienna, Austria) [52] and based on the single nt polymorphisms in the 64 kbp conserved central region of the genome (from 74,472 to 138,348 nt of the genome of ECTV-M) extracted from the whole genome alignment using MEGA5 [53,54]. The phylogeny includes previously published genome sequences of ECTV-M, ECTV-N, and ERPV. Bayesian ancestral state (phylogeography) and tree reconstruction was performed in BEAST v1.10.4 (University of Edinburgh, Edinburgh, UK, and University of California, Los Angeles, CA, USA) [55], using a HKY substitution model with empirical base frequencies and gamma distribution of site-specific rate heterogeneity, a constant size demographic prior, a strict molecular clock, ascertainment bias correction (ABC) model to take into account that only variable sites are being used [56], a discrete trait, asymmetric transition (migration) model, and Bayesian stochastic search variable selection (BSSVS) models. The strength of evidence against the null hypothesis (*H_0_*) was evaluated via MLE comparison with the more complex model (*H_A_*)*_,_* referred to as the Bayes Factor (BF), wherein *ln*BF < 2 indicates no evidence against *H_0_*; 2–6—weak evidence; 6–10—evidence; 10–20—strong evidence, and >20 indicates very strong evidence [57]. Markov chain Monte Carlo (MCMC) samplers were run for 1 billion generations to achieve proper mixing of the Markov chain evaluated by calculating the Effective Sampling Size (ESS) of the parameter estimates with Tracer v1.7 (University of Edinburgh, Edinburgh, UK, and University of California, Los Angeles, CA, USA). ESS > 250 (after 10% a burn-in) were considered robust. The posterior distribution of the sampled trees was summarized within the maximum clade credibility (MCC) tree using TreeAnnotator and specifying a burn-in of 10% and median node heights. The MCC tree was visualized in FigTree v1.4.2 (University of Edinburgh, Edinburgh, UK), and nodes with PP ≥ 0.99 were considered to be evidence of statistically significant phylogenetic relationships. Xml file is available upon request.

## 3. Results

### 3.1. Different Virulence Patterns and Foot Swelling Elicited by ECTV Isolates and Strains in BALB/cJ Mice

ECTV can infect all laboratory mouse strains but has different severities depending on the mouse strain and the route of infection [58]. C57BL/6 and AKR strains are resistant to severe disease by s. c. infection in the footpad, whereas A, BALB/cJ, DBA, and C3H are susceptible to severe disease by this route [59,60]. The virulence of several ECTV strains/isolates has been studied in diverse mouse models [4,13,34,61]. However, due to the diversity of mouse genetic backgrounds used in previous studies, it is difficult to reach clear conclusions about their comparative pathogenesis. We have analyzed the virulence of nine ECTV isolates and strains in susceptible BALB/cJ mice infected s. c. in the footpad, the classical model of mousepox. The set of ECTV isolates tested in this experiment included the virulent isolates ECTV-M [41] and ECTV-H [8], three virus isolates (ECTV-M1, ECTV-M4 and ECTV-M5) from laboratory outbreaks in Central Europe [24] and a natural ECTV isolated from a wild mouse in Germany (ECTV-MK). We also included ECTVs attenuated after serial passage in cell culture (ECTV-I) [28] or CAMs (ECTV-HE and ECTV-MH) [15,16].

Unlike the highly virulent phenotype of ECTV-M with a lethal dose 50 (LD_50_) value bellow 1 pfu (10), mice infected with ECTV-H did not show a dose-dependent mortality and morbidity in two independent experiments despite following a dose-dependent behavior in foot swelling, with similar high degree of swelling occurring at all doses with certain delay, and signs of illness such as hunch back, absence of mobility and conjunctivitis since 4–6 dpi (Figure 1, Appendix A). Thus, for the ECTV-H isolate, it was not possible to estimate a LD_50_ in the BALB/cJ mouse model. Similar experiments performed in DBA/2 mice showed a dose-dependent mortality for ECTV-M and ECTV-N, but not for ECTV-H, and confirmed the attenuated phenotype of ECTV-I (Appendix A).

ECTV-I, ECTV-HE and ECTV-MH were highly attenuated with estimated LD_50_ > 10^6^ pfu, as previously described in other mouse strains [4,16] (Figure 1, Appendix A). The unique death of a mouse infected with 10^6^ pfu of ECTV-I is unlikely to be related to the viral infection since it occurred earlier than the first death of mice infected with virulent strains. Mice infected with ECTV-HE did not show weight loss but instead showed symptoms of illness, including loss of hair in the ocular zone and ulcerative rash in the tail, between 6–14 dpi followed by a slow recovery; whereas mice infected with several doses of ECTV-MH and ECTV-I showed by 15 dpi a small but reproducible weight loss. ECTV-M infection caused a rapid appearance of illness ranging from 8–11 dpi, just before the death of the animals and including ruffled fur, hunched posture and severe ulcerative rash in the tail. In contrast, mice infected with ECTV-MH and ECTV-I showed signs of illness for an extended period of time, ranging from 6 to 19 dpi. Unlike ECTV-M, ECTV-MH and ECTV-I also caused a severe ulcerative rash in ocular area with consequent conjunctivitis-like reaction and loss of fur around the eyes. Moreover, the highly attenuated ECTV-HE, ECTV-MH and ECTV-I strains elicited a severe foot swelling from 4–6 dpi suggestive of immune system activation. This inflammation evolved with the natural amputation of the inoculated foot in all mice infected with ECTV-I at all doses, with the exception of one mouse infected with 10^4^ pfu that suffered an advanced state of necrosis. By contrast, only one mouse infected with 10^6^ pfu of ECTV-MH lost the foot and all animals infected with ECTV-HE recovered from the foot swelling by the end of the experiment. Therefore, these three highly attenuated strains showed clear differences in the way that the local immune system resolves the exacerbated inflammation at the inoculation site.

ECTV-M1, ECTV-M4, ECTV-M5 and ECTV-MK exhibited intermediate estimated LD_50_ values, with ECTV-M1 being the most virulent and ECTV-M4 the most attenuated isolate (Figure 1, Appendix A). Inoculation of 10^2^ pfu of ECTV-M5 caused 60% mortality by 9–10 dpi and the same doses caused 80% mortality in ECTV-MK but a week later. Signs of illness were severe in all animals infected with these four ECTV strains, but weight loss was more pronounced in mice infected with ECTV-M1, the most virulent virus. ECTV-M1, ECTV-M4 and ECTV-M5 elicited a strong footpad swelling with values similar to those recorded for highly attenuated strains by 8 dpi (Figure 1, Appendix A). Consistently, ECTV-M4, the most attenuated virus of the group, caused the same paw strangulation observed in mice infected with ECTV-I and ECTV-MH, without further amputation. Intriguingly, the virus isolated from a wild mouse (ECTV-MK) did not cause swelling of the foot at any time, even in mice infected with high doses.

### 3.2. Only ECTV-H and ECTV-HE Produce ATI Bodies with Embedded Virions

To gain insights into the differences observed in BALB/cJ pathogenesis, we first investigated the phenotype of the ECTV isolates and strains in cell culture. Both attenuated and virulent ECTVs efficiently produced viral plaques in BS-C-1 cells with the same morphology at 3–4 dpi, and exhibited similar growth curves in experiments limited to a one-step growth curve experiment (Figure 2A,B). Interestingly, ECTV-M1 and ECTV-M5 formed syncytia and produced the highest virus yield and fastest growth curves. These results ruled out possible viral replication or propagation impairments as the cause of the attenuated phenotype shown by some ECTVs in vivo.

ECTV forms ATI bodies in infected cells and it has been reported that ECTV-H presents intracellular mature virions either embedded into or on the surface of ATI bodies [28,62]. We confirmed the presence of ATI bodies containing virions (V^+^ phenotype) in 90% of the BS-C-1 cells infected with ECTV-H and ECTV-HE by electron microscopy (Figure 2C). However, we found that this phenotype constitutes an exception within the ECTV isolates and strains tested as the other eight ECTVs, including ECTV-M and ECTV-N, formed ATI bodies with a V^−^ phenotype, i.e., without virions either attached to or embedded into ATIs.

### 3.3. Genomic and Phylogeographic Analyses Unveil Two Major Clades within the ECTV Isolates and Strains

The full-length genome sequence of eight ECTVs was determined by using the 454-Roche deep-sequencing technology (Table 2). The entire coding regions were de novo assembled into two contigs of 35 and 163 kbp separated by the long and variable DRIII region [34]. The DRIII region of the ECTV-MK genome was PCR-amplified as previously described for the other ECTVs [34] and 63 repeats of 24 bp were estimated for ECTV-MK (data not shown). The non-coding terminal regions DRI and DRII were not determined in this study. Consistently with the genome length reported for the three available ECTV genome sequences [34,37,41], the new eight ECTV genomes ranged from 204 to 208 kbp, and the sequence was determined with an average coverage between 33× and 390×. Additionally, the genome of ECTV-M was re-sequenced by using the Illumina technology with an average coverage of 5001× (Table 2). The A+T content of 66.6–66.8% found in the ECTV genomes is characteristic of orthopoxviruses. The central region of the genome of the ECTV isolates and strains was highly conserved and mutations largely accumulated towards the ends of the genome, where the immunomodulatory and virulence genes are located (Figure 3A).

The maximum clade credibility (MCC) tree based on the alignment of 64 kbp from the central region of the ECTV genome showed that ECTV isolates and strains group into two clades that we defined as European clade and Asian clade (Figure 3B). The first clade comprises the European isolates and strains ECTV-H, ECTV-HE, ECTV-M, ECTV-MH, ECTV-M1, ECTV-M4, ECTV-M5 and ECTV-MK, and the Japanese isolate ECTV-I. In agreement with their geographic origin, ECTV-M1, ECTV-M4, ECTV-M5 and ECTV-MK are closely related and form a subclade within the European clade that we named Central-European subclade. These isolates were independently isolated in Germany in 1976 in the cities of Munich (ECTV-M1) and Nunberg (ECTV-M4), in the region of Krefeld (Germany) in 2008 (ECTV-MK), and in Wien (Austria) in 1994 (ECTV-M5) [24,25]. The Asian clade includes ERPV and ECTV-N, whose relationship was previously described [34]. ECTV-H and ECTV-M are the most related ECTV genomes differing in only 30 nt (Table 3). These 30 nt found in ECTV-H are also conserved in the other ECTV genomes belonging to the European clade, suggesting that the European ECTV isolates and strains are more closely related to ECTV-H than to ECTV-M, the reference isolate for ECTV. The European and Asian clades diverged around 177 CE (95% Higher Posterior Density interval [HPD] 772 BCE-877 CE). The time of the most common ancestor (tMRCA) of the Asian clade, composed by ERPV strain form China and ECTV-N from USA [34], is 1960 (95% HPD 1943–1976).

The Bayesian phylogeographic analysis is in agreement with genomic information and the history of the outbreaks. Isolates belonging to the European clade find their origin in the United Kingdom, connected to the early outbreaks caused by ECTV-H (Figure 3B). The time when ECTV-H, progenitor of the European outbreaks, diverged from ECTV-MH, is also the tMRCA of the European clade: 1896 (95% HPD 1866–1918). A clear unidirectional flow of ECTV from the United Kingdom to Germany, to Russia and to Japan was observed (Figure 3C, Table 4). The Central-European subclade emerged in 1942 (tMRCA 95% HPD 1928–1956). Within this clade, the wild mouse isolate ECTV-MK diverged from ECTV-M5 in 1968 (tMRCA 95% HPD 1956–1981), indicating that its ancestor emerged sometime between 1968 and 2008, the year of its first isolation. This analysis corroborates that infection of mouse colonies in Germany in the 1970s may have caused occasional spillover events of a virus closely related ECTV-MP5 into the wild mouse population in Germany, where the natural ECTV-MK isolate was found, and further raising the possibility of the existence of a wildlife reservoir of ECTV in Europe. Strong evidence of viral flow was also observed from Germany to Austria where ECTV-M5 was isolated [25], suggesting again that ECTV was or is circulating within a wildlife reservoir in Germany. The analysis draws a possible two-way connection between China and USA (Figure 3C, Table 4). However, it is conceivable to assume the movement of ECTV to be unidirectional, and specifically from China to USA [34,37]. Finally, this analysis does not provide clear information about evolution direction between European and Asian clades.

The strength of evidence against the null hypothesis (*H*_0_) was evaluated via MLE comparison with the more complex model (*H_A_*), referred to as the Bayes Factor (BF), wherein *ln*BF < 2 indicates no evidence against *H*_0_; 2–6—weak evidence; 6–10—evidence; 10–20—strong evidence, and >20 indicates very strong evidence [57].

### 3.4. ECTV-HE Evolved from ECTV-H through a Genomic Re-Organization

Comparison by dot plot analysis of ECTV-H and ECTV-HE, a strain derived from ECTV-H after serial passages in CAMs, revealed an intra-genomic recombination between the left and right terminal regions (Figure 4A). As a result, a fragment of 10 kbp from the left end of the genome, containing eight genes, had been replaced by a region from the other end of the genome that encompasses three genes that appear now duplicated in ECTV-HE (Figure 3A and Figure 4B). This genomic reorganization likely occurred during the multiple passages of ECTV-H in CAMs and is supported by the presence of several reads in ECTV-HE which contain the repeated terminal region fused to unique sequences from the central region (Figure 4C).

### 3.5. Identification of Putative Virulence and Anti-Inflammatory Proteins

The analysis of truncated genes (including pseudogenes that might not translate into active proteins) among strains that have shown different virulence or pathogenesis in vivo may allow the identification of genes that contribute to the high virulence of some ECTV isolates. That is the case of the highly attenuated ECTV-I that harbored three pseudogenes as compared to the full-length versions present in ECTV-M. We hypothesized that any of these genes might be responsible for its attenuated phenotype in BALB/cJ mice (Figure 5A, Appendix A). In particular, one of these genes, *EVI003P*, has a frame-shift that leads to the loss of 90 amino acid (aa) from the C-terminus of an ankyrin-like protein. The full-length version of the orthologue in ECTV-M (EVM002) inhibits NF-*k*B activation and plays an important role in the pathogenesis of this virus [43]. The ECTV-HE genome, due to the intra-genomic reorganization, lost eight genes that encode for known immunomodulatory proteins such as the viral CD30, viral growth factor, a smallpox virus-encoded chemokine receptor (SECRET) domain-containing protein, a kelch-like protein and two ankyrin like proteins with unknown function (Figure 5A) [41,42,63,64,65,66]. Finally, ECTV-MH genome also contains multiple truncated genes that could account of its attenuation in BALB/cJ mice. Among them, there is an orthologue of EVM001, which encodes for the viral chemokine binding protein 35 kDa, a well-known immunomodulatory protein in VACV [67,68]. The absence of one or several of these genes could account for the highly attenuated phenotype of these three viruses.

The Central-European ECTV-M1, ECTV-M4, ECTV-M5 and ECTV-MK isolates share a common phylogenetic origin. However, ECTV-MK is the only one that does not cause foot swelling. Compared to the gene content of ECTV-M1, ECTV-M4 and ECTV-M5, only four mutated genes of ECTV-MK encode for proteins with aa changes. The orthologue of a kelch-like protein described as a virulence factor for ECTV (EVM018) [69], and the Ank/F-box protein involved in inhibition of NFκB (EVM154) [70] showed only single aa changes. Most importantly, two genes with unknown function (*EVM169* and *EVM11*) encode for proteins with a deletion of 40 residues or five consecutive aa changes, respectively (Figure 5A, Appendix A). Some of these genes might be responsible for the early control of the inflammatory process during ECTV-MK infections.

### 3.6. The Genome of ECTV-H Strain Encodes a Full-Length P4c Protein Required for Viral Inclusion into the ATIs

Re-sequencing of the ECTV-M genome with the Illumina technology showed that nt at positions 39,775 and 102,467 are erroneous in the previous sequence published by Chen et al. [41] and therefore no difference was considered to be existing at those positions between the genomes of ECTV-M and ECTV-H (Table 5). However, we have found that ECTV-H differs from ECTV-M in 30 nt, which are distributed in nine positions along the genome (Table 3 and Table 5), affecting the coding region of only four genes: (i) the *EVH041* gene, orthologue of *EVM025*, differs in the number of DIDNGIVQ repeats within the highly variable DRIII region, as previously described for all ECTV isolates (Figure 5A) [34]; (ii) the *EVH012* orthologue of *EVM008*; (iii) the *EVH44* orthologue of *EVM028*, shows a single nt change that translates into a conservative aa change (Table 5); and (iv) the *EVH148* gene has a single nt change that eliminates a stop codon present in the pseudogene Region Q from the ECTV-M genome, coding for a truncated protein of 183 aa, and extends the protein encoded by *EVH148* to 503 aa (Table 4). This protein corresponds to the ECTV orthologue of the VACV P4c protein (VACWR149) involved in the inclusion of viral particles into ATI bodies [71] (Figure 5A,B). A full-length p4c protein would be expressed by ECTV-H and ECTV-HE, whereas the other ECTV isolates and strains encode truncated versions of the P4c protein that lack the domain needed to direct the virions towards ATI bodies (Figure 5B). This result provides the genetic basis for the different V^-^ and V^+^ phenotypes previously observed in vitro among ECTV isolates and strains, since only those viruses encoding full-length versions of the P4c protein include their virions into ATIs. Consistently with the V^−^ and V^+^ phenotypes observed *in vitro* spleen cells from ECTV-H-infected mice showed intracellular mature virions occluded inside or in the periphery of ATI bodies (V^+^ phenotype) and ATI bodies from spleen cells of mice infected with ECTV-M showed a V^−^ phenotype (Figure 5C).

### 3.7. In Vivo Dissemination and Transmission of ECTV-M Is More Efficient Than ECTV-H

The role that ATI bodies play in the life cycle of many poxviruses is unclear [28,72,73], but it has been suggested that they might be involved in viral transmission or dissemination processes [74,75,76,77,78,79]. The predicted functional differences in ECTV-H as compared to ECTV-M are restricted to two conservative amino acid changes, an insertion of a repeated amino acid motif in EVM025 and the expression of a full-length p4b protein (Table 5, Appendix A). We took advantage of the fact that a full-length 4b protein was the main difference between ECTV-H and ECTV-M to investigate the hypothesis that ATI bodies may play a role in viral dissemination within the infected host. We analyzed the viral load in popliteal lymph nodes, spleen and liver from mice infected s. c. in the footpad with 10 pfu of ECTV-H or ECTV-M (Figure 6A). At 3 dpi, both ECTV-H and ECTV-M replicated to high viral titer in popliteal lymph nodes, 5 × 10^5^ pfu/g and 6 × 10^6^ pfu/g, respectively. At 5 dpi, ECTV-M replication in the spleen and liver was significantly higher than the level of replication of ECTV-H, suggesting more-efficient intra-host dissemination when virions are not embedded in the ATIs. At 7 dpi, both viruses replicated to similar levels in spleen and liver.

We also explored the role of ATI bodies in viral inter-host transmission by studying the ability of ECTV-M and ECTV-H to be transmitted from infected mice to naïve mice housed in the same cage (Figure 6B). Transmission was not detected from infected mice in the early days of infection (2 to 5 dpi, group 1) (Figure 6C). Interestingly, ECTV-M transmission from infected mice was significantly more efficient than transmission of ECTV-H during the period of 4–7 dpi (group 2) (Figure 6C). Five out of seven naïve mice exposed to ECTV-M-infected mice showed high titers in the spleen and the infection was lethal in 75% of the animals by 22 days post-exposure. By contrast, only one mouse exposed to ECTV-H-infected mice developed a detectable infection and no mortality was observed. Both viruses were efficiently transmitted to naïve mice between 6 and 9 dpi (group 3) (Figure 6C).

## 4. Discussion

Our study reports the genomic and phylogeographic analysis of a collection of ECTV isolates and attenuated variants. Our data is in agreement with the hypothesis that the constant exchange of biologic material, such as cells, tissues and serum, from laboratories in UK and laboratories in Europe, Japan and USA caused the first wave of ECTV outbreaks between the 1930s and the 1980s, whereas the most recent outbreaks in USA were caused by an independent introduction of mouse serum from China [34]. The V^+^ phenotype of ECTV-NIH79, a strain responsible for an outbreak in USA in the 1980s, is in line with the hypothesis that the early USA outbreaks were caused by material contaminated with ECTV-H, the only isolate with this V^+^ phenotype [4]. Similarly, our Bayesian phylogeographic analysis suggests that the introduction of ECTV in Russia might have been caused by material contaminated with ECTV-H. ECTV-M and ECTV-H differ only in 30 nt mutations presumably introduced in ECTV-M and not found in other ECTV isolates from the European clade. Hence, we propose that these mutations might have been acquired by ECTV-M during subsequent cell culture amplification of the original ECTV-H isolate. The origin of ECTV-MH is uncertain. Unlike all other ECTVs, ECTV-MH presents a full-length *EVMH010* gene that encodes a C-type lectin suggesting an ancient origin of this virus within the species. However, we cannot rule out the possibility that ECTV-MH, like ECTV-HE, was originated after extensive passage of ECTV-H, or a closely related virus, in CAMs. Moreover, ECTV-MH may be related to ECTV-HE since these viruses share the same mutations introduced in many genes as compared with ECTV-H, including the restoration of the ECTV-M pseudogene Region N (full-length gene *EVN038* in ECTV-N). Similarly, the genomic sequence of ECTV-I is consistent with this strain being derived from ECTV-H after serial tissue culture passages and the introduction of mutations. This strain is the most closely related to ECTV-H among the attenuated strains, differing only in 138 nt. Our Bayesian phylogeographic analysis evidenced that the importation of ECTV in Japan likely occurred from the UK.

The genomic sequence and virulence of ECTV-MK in a susceptible mouse model provides unprecedented information on ECTV natural infections. The finding of ECTV-MK infecting a wild mouse in the same European region where ECTV-M1, ECTV-M4 and ECTV-M5 were isolated, suggests that direct contact of laboratory and wild mice infected with ECTV-M1/-M4/-M5/MK-like viruses might have caused these outbreaks [24,25]. These isolates have not been passed extensively in cell culture, and therefore their genomes should be very close to the natural circulating viruses. In 1962, a mousepox record in wild mice in Germany was reported [4,39]. However, the presence of ECTV was not confirmed by genomic sequencing and therefore there is no definitive demonstration that ECTV caused that infection in wild mice [4,39]. The existence of a recombinant virus of ECTV and CPXV, CPXV-No-H2, isolated from a human patient in Norway, was reported to have ECTV sequences as a result of recombination events between CPXV and ECTV. This raised the possibility of a reservoir for ECTV in that country, although no evidence of infections in wild mice were reported [80]. We therefore report the first demonstration of ECTV isolation from wild mice, in the region of Krefeld in Germany in 2008. As it is known, ECTV infects susceptible and resistant laboratory mouse strains (*M. musculus*) and these strains result from the genetic interchange of wild mice strains from Europe and Asia [13,81]. The wild mouse species infected with ECTV-MK is unknown, and the most probable candidates are *M. musculus domesticus* or *M. musculus musculus,* which are common in central Europe and showed moderate resistance to ECTV-NIH79 [82]. The existence of mice infected with ECTV in China has been reported since the 1960s, although the natural source remains unknown [34,37,83]. The split between ECTVs from the European and Asiatic clades dates back to 177 CE and reflects the large genetic distance between the two clades, possibly a consequence of the long adaptation process to geographically and genetically different mouse strains [84].

Our study constitutes the first comprehensive evaluation of the in vivo phenotype of a collection of ECTV isolates and strains in the same mouse model. The potential influence of mouse sex in the susceptibility to ECTV infection should be addressed in the future, although to our knowledge there is no evidence for a major effect of sex on ECTV susceptibility in this mousepox model. The integration of the full genome sequence information of the ECTVs with their pathogenesis in the classical mousepox model of susceptible BALB/CJ mice and replicative properties in tissue culture led us to identify a set of putative host-interaction genes including molecular determinants of virulence. A candidate gene for inducing fusion of BS-C-1 infected cells leading to a faster in vitro replication of ECTV-M5 and ECTV-M1 is *EVM5-115*/*EVM1-115*. These genes share sequence differences with the corresponding genes of the related ECTV-M4 or ECTV-MK that exhibit a slower replication cycle and encode for an orthologue of the VACV Copenhagen strain *D8* gene [85]. The D8 protein shows sequence similarity to a cellular carbonic anhydrase (EC 4.2.1.1), and the mutations Y69H and Y92H increase the activity of the protein and viral replication in vitro [86,87]. The proteins encoded by *EVM1-115* and *EVM5-115* show mutations (H67R and S206- in EVM1-115; N175K in EVM5-115) that could account for enhancement of its carbonic anhydrase and the subsequent decrease of the pH of the cell facilitating cell fusion mediated by A17 and A27, and ultimately boosting of viral replication.

Similarly to other CAM-adapted strains such as Chorioallantois VACV Ankara and modified VACV Ankara, we reported the loss of several genes encoding for ankyrin-like or kelch proteins as responsible for the attenuation of ECTV-MH and ECTV-HE [69,88,89,90,91,92,93,94]. The former encodes also for a truncated version of the viral chemokine binding protein 35K [16], whose ortholog in VACV reduces host inflammation in the lung of infected mice [68]. A candidate gene responsible for the ECTV-I attenuation and strong inflammatory response at the site of infection is *EVM002*, since this is the only gene of ECTV-I truncated in comparison with other virulent ECTVs and it has been involved in the blockade of NF-*k*B [43]. However, the milder inflammation observed in mice infected with ECTV-MH as compared to ECTV-I suggested that *EVM002* is a stronger anti-inflammatory mediator than *EVM001* and other genes truncated of ECTV-MH. The absence of a functional protein encoded by *EVM002* is likely involved in the characteristic amputation that named the virus “ectromelia” in ECTV-I. This amputation had been previously described in three-week old mice infected with ECTV-H [39]. The older age of the mice used in our experiments might explain the absence of amputation observed in ECTV-H. It would be of interest to further investigate whether the attenuated phenotype is due to loss of function of one or more than one of these candidate genes.

ECTV virulent strains encode several anti-inflammatory proteins that prevent local swelling a few days after infection such as CrmD or the type I interferon binding protein [43,45,46,64,95,96]. ECTV isolates from the Central European subclade (ECTV-M1, ECTV-M4 and ECTV-M5) infections elicited an early footpad swelling even in animals infected with lethal doses. Interestingly, the genetically and geographically related ECTV-MK, isolated from a wild mouse, exhibited a complete blockade of footpad swelling. By comparing their gene content, we identified two genes with novel putative anti-inflammatory properties: *EVMK017* (*EVM011*) and *EVMK199* (*EVM169*). *EVMK017* encodes an orthologue of VACV Copenhagen strain C10, a protein that has a region between aa 316 and 327 with sequence similarity to the IL-1 receptor antagonist, and it has been suggested, although not experimentally demonstrated, that this region of C10 may interact with the IL-1 receptor and block its function [97]. The EVMK017 protein presents a cluster of five aa changes in positions that correspond to this C10 region, suggesting that this protein, if demonstrated that it interacts with the IL-1 receptor, could be involved in the inhibition of the IL-1 signaling pathway [97]. The second gene, *EVMK199*, encodes the largest orthopoxvirus protein [98], a protein shown to block T cell activation and to contribute to virulence [99,100]. EVMK199 shows a 40 aa deletion in the same region where its homologue in ECTV-M5 has seven nearly consecutive aa changes. This might explain why ECTV-M5 also prevented inflammation at low dosages. The implication of these genes in poxvirus pathology is evidenced by the attenuated phenotype observed in fowlpox strains with truncated versions of the orthologous genes of *EVMK017* and *EVMK199* [88].

We also addressed the virulence exhibited by ECTV-H and ECTV-M during infection of BALB/cJ mice and a dose-independence lethality of the former despite sharing almost identical genomes. Based on their genome sequences and on in vitro and in vivo phenotypes, we hypothesize that the inclusion of ECTV-H virions into the ATIs, mediated by the full-length version of the P4c protein encoded by the *EVH148* gene, may be responsible for the unpredictable virulence phenotype displayed by ECTV-H. The P4c protein plays a major role in the process of occlusion of the virions into the ATI bodies [75]. ECTV-H and ECTV-M replicate in mice to similar levels early after infection in lymph nodes. The truncation of the gene encoding P4c in ECTV-M, and in all the other ECTV isolates and strains, may accelerate the viral intra-host dissemination and enhance viral transmission among hosts, whereas the presence of a P4c full-length protein in ECTV-H directs virions to ATI bodies and may reduce their ability to disseminate within the mouse and transmit to other mice.

In contrast to the ECTV isolated after the ECTV-H outbreak of 1930 in London, which has a V^+^ phenotype, the ECTVs that were transferred from this initial source to other laboratories, such as ECTV-M or ECTV-I, have a V^-^ phenotype. The selection of a V^-^ phenotype may be the consequence of a more rapid replication and dissemination in cell culture and mice of the V^-^ variants [75,79]. Consistent with this, the three isolates from the Central European subclade (ECTV-M1, ECTV-M4 and ECTV-M5) all have a V^−^ phenotype that may have facilitated transmission to mouse colonies in animal house facilities. The possibility that poxviruses occluded in ATI bodies and incorporated by phagocytic cells may escape the lysosome degradation pathway exploiting the A17-A27 fusion activity at low pH suggests that embedding of virus particles into ATI bodies may facilitate a non-specific mechanism of viral entry [76,85]. Accordingly, CPXV shows broad host range and presents a V^+^ phenotype, while VARV and ECTV are host-specialized viruses and exhibit V^−^ phenotype, except for ECTV-H [101]. As classical host-range studies have not been performed for ECTV, we do not know whether ATI bodies may actually represent a non-specific mechanism of viral entry and broaden ECTV-H’s host range. It remains to be explored whether the frequently reported hypothesis that occlusion of virions into ATI bodies provide some other advantage related to the physical resistance of virions out of the animals and a long-term transmission of the virus. Consistently with the virulence phenotype of ECTV-M and ECTV-H, CPXV lacking the ability to occlude into ATI bodies is more virulent than the variant that does occlude into ATI bodies [79]. Therefore, ATI bodies might be a viral self-contention mechanism that ensures host survival and persistence in natural reservoirs.

In summary, this is the first comprehensive report that sheds light into the phylogenetic relationship of the worldwide outbreaks caused by ECTV, and we show that the first ECTV isolated from a wild mouse, ECTV-MK, is related to a Central European subclade. Genome sequencing and pathogenesis studies have identified ECTV genes that may contribute to virulence or control the host inflammatory response. We show that most of the ECTV isolates and strains cannot recruit virions to ATIs, and this may enhance their transmission efficiency.

## Figures and Tables

**Figure 1 viruses-13-01146-f001:**
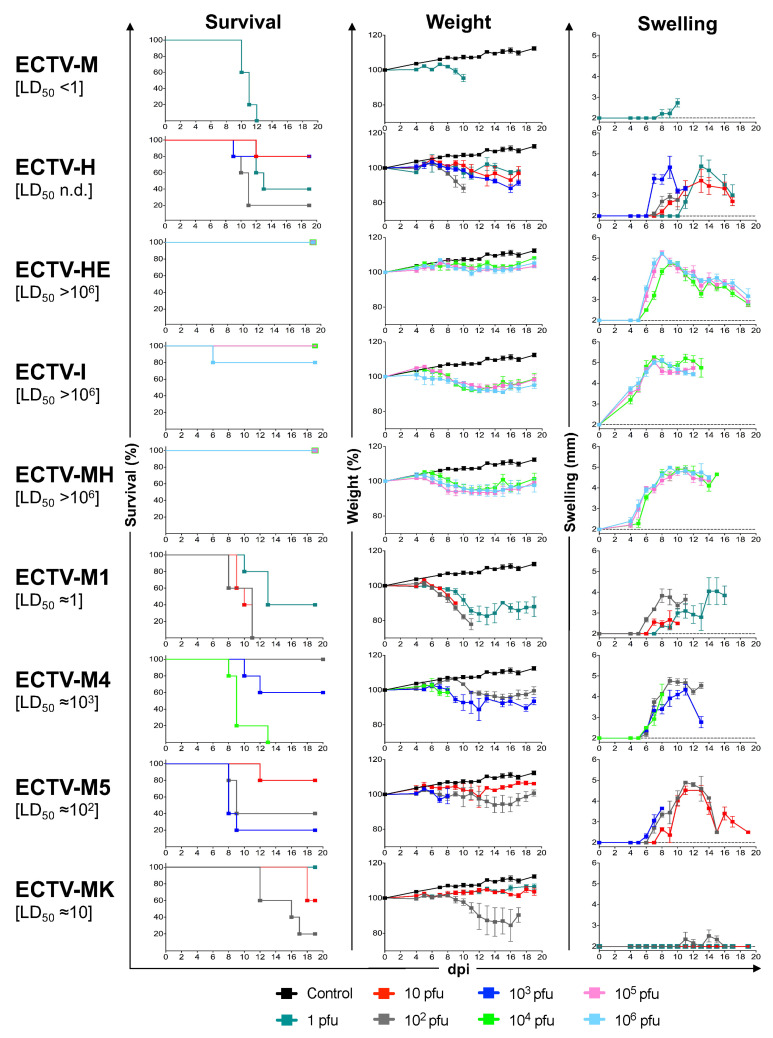
Virulence of ECTV isolates and strains in susceptible mice. Groups of five female BALB/cJ mice were infected s. c. in the footpad with the indicated ECTV and monitored during 19 d. The percentage of survival and weight, with respect to uninfected mice, and swelling of the inoculated foot are shown from left to right. Mice were infected with various doses: 1 pfu (dark green line in ECTV-M, ECTV-H, ECTV-M1, and ECTV-MK), 10 pfu (red line in ECTV-H, ECTV-M1, ECTV-M5, and ECTV-MK), 10^2^ pfu (gray line in ECTV-H, ECTV-M1, ECTV-M4, ECTV-M5, and ECTV-MK), 10^3^ pfu (dark blue line in ECTV-H, ECTV-M4, and ECTV-M5), 10^4^ pfu (light green line in ECTV-HE, ECTV-I, ECTV-MH, and ECTV-M4), 10^5^ pfu (pink line in ECTV-HE, ECTV-I, and ECTV-MH) or 10^6^ pfu (light blue line in ECTV-HE, ECTV-I, and ECTV-MH). The black line represents control animals. Mean ± SD are represented. M, ECTV-M; H, ECTV-H; HE, ECTV-HE; I, ECTV-I; MH, ECTV-MH; M1, ECTV-M1; M4, ECTV-M4; M5, ECTV-M5; MK, ECTV-MK. The results shown with ECTV-MP1, ECTV-MP4, ECTV-MP5, ECTV-MK and ECTV-H are representative of two experiments.

**Figure 2 viruses-13-01146-f002:**
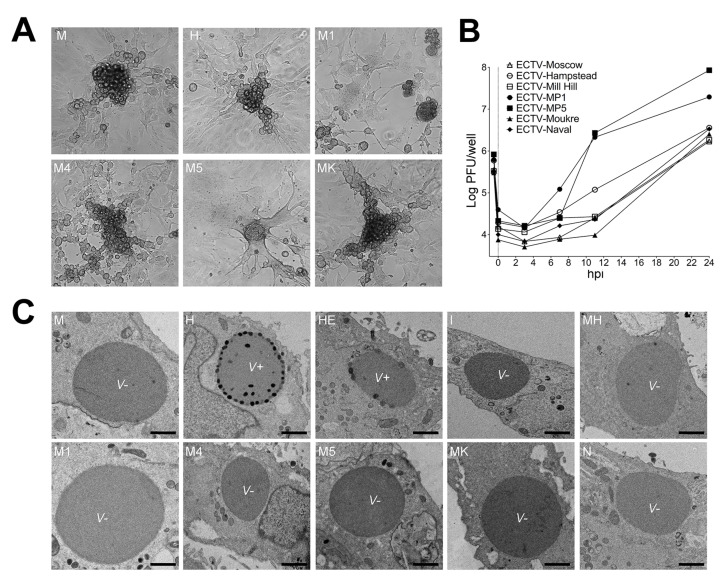
Replication of ECTV isolates and strains in tissue culture. (**A**) Representative micrographs of lysis plaque morphology formed by the indicated ECTV isolate/strain in BS-C-1 cells at 96 hpi were captured with a phase contrast microscope Leica DM IL LED (Leica Microsystems) at 400× magnification. (**B**) One-step growth curve experiment of ECTV infection. BS-C-1 cells were infected with 3.5–10 pfu/cell of the indicated ECTV isolate/strain and virus titers determined at different times after infection. The results are representative of two experiments and each point is the mean of two lysis plaque assay determinations. Virus titer is indicated in Log10. (**C**) Representative electron micrographs of ECTV-induced ATI bodies in BS-C-I cells at 24 hpi. The presence of virions in ATI bodies is indicated with V^+^ and the absence of virions with V^−^. Bars: 400 nm.

**Figure 3 viruses-13-01146-f003:**
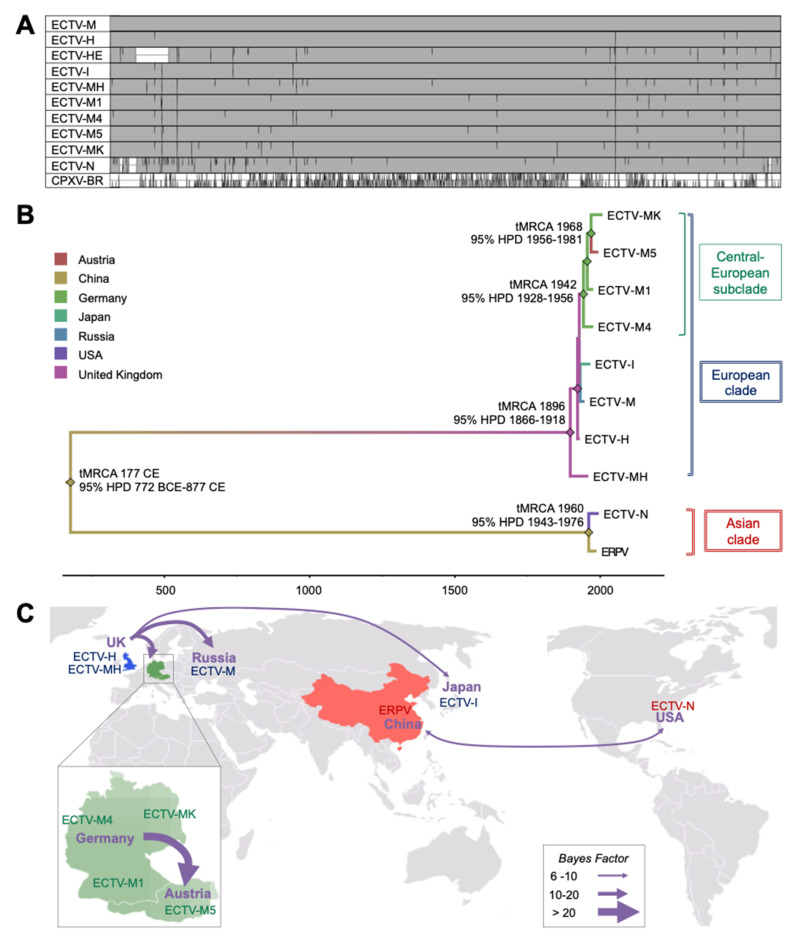
Genomic comparison and phylogenetic analysis of ECTV isolates and strains. (**A**) Genome alignment of ECTVs to the reference genome of ECTV-M, showing 100 bp windows with identities below 99%. Vertical lines indicate single nt changes as compared to ECTV-M. The white box in ECTV-HE indicates the absence of that particular genomic region as compared to the other strains. In this analysis, regions containing direct sets of repeated sequences, as for the DRI and DRII regions (1–1548; 208,224–209,771 in ECTV-M genome) and the DRIII (reduced to an arbitrary number of 15 repeats for all strains), were not used for the comparison. The low genomic similarity between the ECTV-M and CPXV-Brighton Red (CPXV-BR) is shown to emphasize the high genome similarity among the ECTV isolates and strains. (**B**) Maximum clade credibility (MCC) tree of 64 kbp of the central region of the genome of all ECTVs whose genome is shown. Branches are colored based on country of origin, diamonds indicate PP > 0.9 and their color indicates the country of origin for the time most common ancestor (tMRCA). (**C**) The map shows the origin and the spread of ECTV causing outbreaks in laboratory mouse colonies worldwide based on Bayesian phylogeographic analysis. Arrows represent the spread of ECTV among countries and the color is based on BSSVS values for the movement. Enlarged parts of the map show the main ECTV reservoirs in the European continent where ECTV-M1-M4-M5 and MK strains have been circulating.

**Figure 4 viruses-13-01146-f004:**
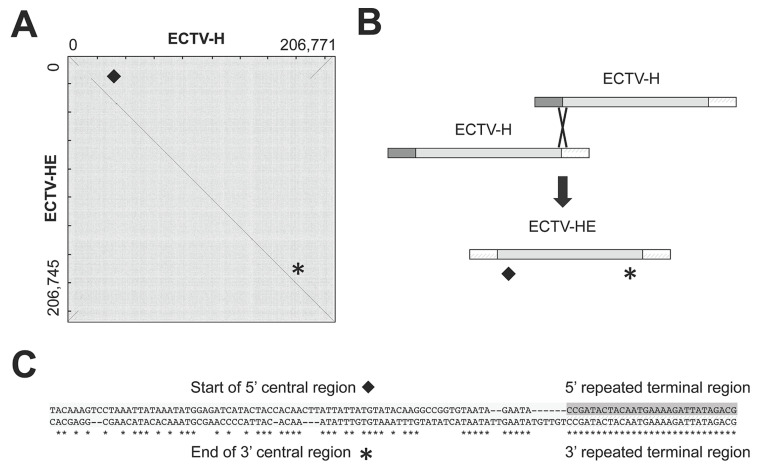
Genomic comparison of ECTV-H and ECTV-HE. (**A**) Dot plot of ECTV-H vs. ECTV-HE. The dot plot compares two sequences by organizing one sequence on the x-axis, and another on the y-axis, of a plot. When the nt of both sequences match at the same location on the plot, a dot is drawn at the corresponding position. (**B**) Schematic representation of the proposed genomic recombination between two ECTV-H genomes to generate ECTV-HE. (**C**) Alignment of two representative reads that share the end of the repeated terminal region of ECTV-HE and differ in the beginning of the central region of the genome. An asterisk and a diamond indicate the approximated region where the aligned reads derived.

**Figure 5 viruses-13-01146-f005:**
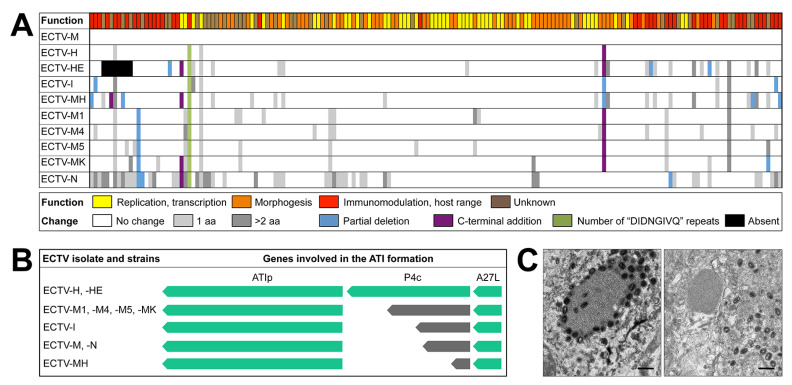
Variability among the proteins encoded by the genomes of the ECTV isolates and strains. (**A**) Comparison of the protein content: each rectangle represents a protein with aa changes (due to non-synonymous nt mutations) when compared to that of the reference ECTV-M. The color of the upper row rectangles indicates the function of each protein (Function) and the lower row indicates the type of modification (Change). Non-synonymous nt mutations that cause termination of the reading frame are indicated by the light blue color (partial deletion). (**B**) ECTV genes involved in ATI bodies morphogenesis and virion inclusion process: genes are shown in green and pseudogenes in grey. (**C**) Electronic micrographs of spleens of mice infected with 10^3^ pfu per mice of ECTV-H, (left), and ECTV-M, (right), at 4 dpi.

**Figure 6 viruses-13-01146-f006:**
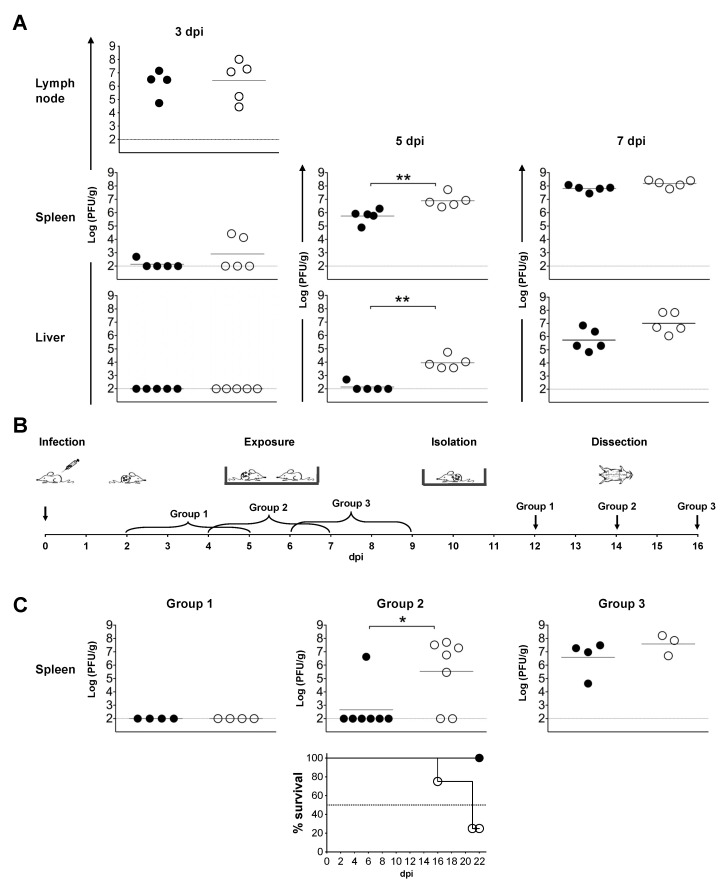
ECTV inclusion into ATI bodies affects its dissemination and transmission. (**A**) Viral dissemination of ECTV-H (●) and ECTV-M (○) from female BALB/cJ mice s. c. infected in the footpad with 10 pfu. Groups of five ECTV-infected mice were euthanized and virus titer in spleen and liver samples collected at 3, 5 and 7 dpi, and virus in popliteal lymph nodes at 3 dpi, are shown. (**B**) Schematic representation of the in vivo viral transmission experiment: two infected mice at 2, 4 and 6 dpi were housed with four naïve mice during 3 days. Animals were subsequently separated and 7 days post-exposure (dpe) euthanized to collect spleen samples. Infected and recipient naïve mice shared the same cage during “Exposure”, then naïve mice were separated from infected animals (“Isolation”) and the viral load was analyzed in the spleen of the naïve mice 7 dpe. Three different protocols of transmission were carried out as schematically shown in the diagrams below the graphs: group 1 of naïve mice was housed with 2–5 dpi infected mice, group 2 with 4–7 dpi infected mice and group 3 with 6–9 dpi infected mice. (**C**) Viral transmission from ECTV-M and ECTV-H s. c. infected (10^3^ pfu) BALB/cJ mice to naïve mice after 3 days of exposure in the same cage. Spleen viral titers are shown at 7 dpe. Viral titers and mortality of group 2 are the result of two independent experiments with 4 and 3 naïve mice, respectively. Statistical significance is shown with * (*p* < 0.05), ** (*p* < 0.01). Black horizontal lines show geometric means. Gray horizontal line indicates the limit of detection.

**Table 1 viruses-13-01146-t001:** ECTV isolates and strains used in this study.

Isolate	Location of Collection	Date of Collection	Origin	Plaque Isolation	Reference
ECTV-Hampstead (ECTV-H)	National Institute for Medical Research, London, United Kingdom	1930	Mouse colony outbreak	no	[8]
ECTV-Moscow (ECTV-M)	Moscow, Russia	1946	Mouse colony outbreak and cell culture passages	yes	[17]
ECTV-Hamsptead Egg(ECTV-HE)	-	1949	ECTV-H passed in chorioalantoic membranes	no	[15]
ECTV-Mill Hill (ECTV-MH)	-	1959	ECTV-H passed in chorioalantoic membranes	no	[16]
ECTV-Ishibashi (ECTV-I)	Ishibashi, Japan	1966	Mouse colony outbreak and cell culture passages	yes	[28]
ECTV-MP1 (ECTV-M1)	Munich, Germany	1976	Mouse colony outbreak	no	[24]
ECTV-MP4 (ECTV-M4)	Nuremberg, Germany	1976	Mouse colony outbreak	no	[25]
ECTV-MP5 (ECTV-M5)	Wien, Austria	1994	Mouse colony outbreak	no	[25]
ECTV-Naval (ECTV-N)	US Naval Medical Research Institute in Bethesda	1995	Mouse colony outbreak and cell culture passages	yes	[35]
ECTV-MouKre (ECTV-MK)	Krefeld, Germany	2008	Wildlife	no	This study

**Table 2 viruses-13-01146-t002:** Summary of data obtained by next-generation sequencing of the ECTV genomes.

ECTV ^a^	Reads	Average Size (bp)	Mapped Reads	Coverage (x)	Genome Size (bp)
ECTV-H	96,434	365	69,503 (72.0%)	122	206,771
ECTV-HE	240,573	370	218,157 (90.7%)	390	206,745
ECTV-I	25,360	396	17,411 (68.6%)	33	207,479
ECTV-MH	120,182	387	117,070 (97.4%)	218	207,108
ECTV-M1	79,430	396	69,600 (87.6%)	133	206,842
ECTV-M4	77,287	385	71,527 (92.5%)	132	207,700
ECTV-M5	46,469	393	41,334 (88.9%)	78	207,348
ECTV-MK	112,670	302	57,779 (51.3%)	83	209,592
ECTV-M	22,391,099	100	10,490,202 (46.85%)	5001	209,771

^a^ All strains were sequenced with 454-Roche technology except the ECTV-M that was sequenced with Illumina technology.

**Table 3 viruses-13-01146-t003:** Number of nt differences among ECTV genomes.

	ECTV-M	ECTV-H	ECTV-I	ECTV-MH	ECTV-M1	ECTV-M4	ECTV-M5	ECTV-MK
ECTV-M	-	30	167	361	148	186	362	589
ECTV-H	30	-	138	331	138	156	353	559
ECTV-I	167	138	-	437	258	274	473	689
ECTV-MH	361	331	437	-	352	348	531	732
ECTV-M1	148	138	258	352	-	130	297	527
ECTV-M4	186	156	274	348	130	-	363	565
ECTV-M5	362	353	473	531	297	363	-	285
ECTV-MK	589	559	689	732	527	565	285	-

**Table 4 viruses-13-01146-t004:** Bayesian phylogeographic analysis.

Transition	Bayes Factor	Evidence against *H*_0_
Germany to Austria	460.2	very strong evidence
United Kingdom to Russia	17.1	strong evidence
United Kingdom to Germany	14.5	strong evidence
China to USA	9.9	evidence
United Kingdom to Japan	7.5	evidence
USA to China	7.2	evidence
Russia to Japan	4.1	weak evidence
USA to United Kingdom	3.2	weak evidence

**Table 5 viruses-13-01146-t005:** Differences at the nt and aa level between the ECTV-M and ECTV-H genomes.

ECTV-M				ECTV-H		aa Change
nt	ORF	Published ^a^	Re-Sequenced ^b^	ORF	Sequenced	
13,526	*EVM008*	A	A	*EVH012*	G	V164A
39,775	*EVM025*	A	C	*EVH041*	C	
41,907	*EVM028*	T	T	*EVH044*	C	I258V
59,134	Intergenic	ATAAGATAAG	-	Intergenic	-	
102,467	*EVM089*	T	A	*EVH107*	A	
147,842	Region Q	T	-	*EVH148*	G	Stop183G
157,777	Region S	21xC	-	EVH167P	10xC	
189,000	Region Y	T	-	EVH195P	TAT	
191,424	Region Y	A	-	EVH195P	AT	
195,294	Region Z	G	-	EVH198P	T	

^a^ Chen et al. [42] ^b^ Sequenced in this study with Illumina technology.

## Data Availability

The genome of ECTV-H was submitted to GenBank with the accession number KY554976.1, and ECTV-HE, ECTV-MH, ECTV-I, ECTV-M1, ECTV-M4, ECTV-M5 and ECTV-MK were deposited to the European Nucleotide Archive (ENA) with accession number PRJEB44436.

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
