# Peer review of "Comparative Pathogenesis, Genomics and Phylogeography of Mousepox"

_viruses, 2021, doi:10.3390/v13061146_

Round 1

Reviewer 1 Report

The authors do a comparative study covering different variants of ectomelia virus (ECTV) isolated from wild mice, at institutional outbreaks, and after serial passage in culture.   Data presented highlight a number of mutations and deletions that may play a significant role in altered virulence observed during infection in a ‘susceptible’ strain of mice.  The authors demonstrate an interesting pattern of geographic transition for the different ECTV variants through phylogeographic analysis.  Lastly the authors highlight a potential role for truncated P4c protein to prevent virion recruitment to A-type inclusion bodies, which ultimately improves viral dissemination.  Taken together, this manuscript is a comprehensive compare and contrast of multiple ECTV variants across multiple geographical landscapes.

Major Comments

-It’s very difficult to keep all the variants straight.  It would be great to have a table (or add to a previous table) simply indicating the source of each variant so readers can quickly find this information when described throughout the manuscript.

-While I understand that more in-depth studies evaluating each gene alteration would be very time consuming and may not work, I would like to know if there are any commonly shared mutations/deletions between ECTV variants HE, I, and MH with ‘poor’ virulence as compared to the more virulent variants.  Particularly any that are in important genes or segments with previously described roles.

-I am concerned about the reproducibility in mouse experiments (Figure 1 and 6).  Have these experiments at least been repeated again? 

-The statements about ECTV-MK and ECTV-H within the abstract are very bold and suggest clear causative effects.  While the changes are observed, there is no experiments done that directly test these statements (adding/removing the inhibition segments from ECTV-ME; add longer P4c in ECTV-M).  Therefore, I believe the statements need to be modified to indicate that these may be the reasons.

Minor Comments:

Introduction

-For readers who are unfamiliar, I believe it would be good to briefly describe how virulence/pathology is greatly altered by mouse strain.    

Materials and Methods

-Use standard nomenclature for Balb/c mouse when describing source

-Provide more details for plaque assay in methods

-I believe it would help to have another column in Table 1 briefly indicating source of each strain for clarity

Results

-Line 232: Data not shown is discouraged in Viruses and should be included as supplemental.

-Figure 1: Any statistical differences for weight loss (Please include)?  Please include that statistical analysis was at least done if so.

-Figure 1: It would be good to include some table or even a number on the swelling graph indicating how many mice lost feet since you mention it. 

-Figure 2: Please indicate what log this is for virus titer.  It is also unclear what the authors mean by a single infection cycle.  Please clarify.

-Figure 4: It is unclear what is being presented in 4A.

-Line 462: Authors refer to popliteal ganglia?  Do you mean lymph node?  Please show data in Figure 6.

-Figure 6: B and C labels are swapped.  Please include survival data in figure.

Author Response

Reviewer 1

The authors do a comparative study covering different variants of ectromelia virus (ECTV) isolated from wild mice, at institutional outbreaks, and after serial passage in culture.   Data presented highlight a number of mutations and deletions that may play a significant role in altered virulence observed during infection in a ‘susceptible’ strain of mice.  The authors demonstrate an interesting pattern of geographic transition for the different ECTV variants through phylogeographic analysis.  Lastly the authors highlight a potential role for truncated P4c protein to prevent virion recruitment to A-type inclusion bodies, which ultimately improves viral dissemination.  Taken together, this manuscript is a comprehensive compare and contrast of multiple ECTV variants across multiple geographical landscapes.

Major Comments

-It’s very difficult to keep all the variants straight.  It would be great to have a table (or add to a previous table) simply indicating the source of each variant so readers can quickly find this information when described throughout the manuscript.

Response: We agree with the reviewer and added a Table to the manuscript (new Table 1).

-While I understand that more in-depth studies evaluating each gene alteration would be very time consuming and may not work, I would like to know if there are any commonly shared mutations/deletions between ECTV variants HE, I, and MH with ‘poor’ virulence as compared to the more virulent variants.  Particularly any that are in important genes or segments with previously described roles.

Response: We agree with the reviewer on the interest of this analysis. The new Table S3 shows changes identified in the genes from all ECTV isolates and strains sequenced, and this Table helps to identify common mutations/deletions in the three ECTVs showing a clear attenuated phenotype. Looking into details at Table S3 it becomes clear that finding common mutations in the genomes of attenuated viruses is not evident. For example, ECTV-HE has a large deletion of 10 genes present in ECTV-H (EVH008-EVH017), and all, but three (pseudogenes), of these genes are present in ECTV-I and ECTV-MH. Similarly, EVHE033 is mutated in ECTV-HE, but this gene is presumably active in both ECTV-I and ECTV-MH. The contrary situation can be seen with gene EVH148, active in ECTV-HE but inactivated in ECTV-I and ECTV-MH. Since it is difficult to draw conclusions from this analysis, we prefer not to include it, and believe that the addition of Table S3 will help the reader to do this analysis if he/she is interested.

-I am concerned about the reproducibility in mouse experiments (Figure 1 and 6).  Have these experiments at least been repeated again? 

Response: The results shown in the manuscript for Figure 1 is the output of one in vivo experiment where all the isolates were analyzed in parallel. The virulence of ECTV-M, ECTV-H and attenuated viruses (ECTV-HE, ECTV-MH and ECTV-I) are well known and described in previous publications. We have repeated the in vivo experiments for the virus isolates that we describe here for the first time: ECTV-MP1, ECTV-MP4, ECTV-MP5 and ECTV-MK, and ECTV-H as a control. The virulence and pathology caused by these isolates were comparable to the result shown in Figure 1. We decided to present the experiment shown in Figure 1 because all viruses were compared in parallel. We added a sentence at the end of Figure 1 legend to indicate that ‘The results with ECTV-MP1, ECTV-MP4, ECTV-MP5, ECTV-MK and ECTV-H are representative of two experiments’. In addition, we add a new Supplementary Table (Table S2) where we compare ECTV-M, ECTV-N, ECTV-H and ECTV-I in DBA/2 mice.

 The results shown in the manuscript for Figure 6 correspond to one or two experiments. All combinations (groups 1, 2 and 3, see Fig. 6B) were include in the first experiment. Since we observed a significant difference between ECTV-H and ECTV-M in group 2, we repeated a second experiment with only these conditions, and found the same results. The Figure 6 legend states that ‘Viral titers of group 2 are the result of two independent experiments with 4 and 3 naïve mice, respectively.’

-The statements about ECTV-MK and ECTV-H within the abstract are very bold and suggest clear causative effects.  While the changes are observed, there is no experiments done that directly test these statements (adding/removing the inhibition segments from ECTV-ME; add longer P4c in ECTV-M).  Therefore, I believe the statements need to be modified to indicate that these may be the reasons.

Response: We have toned down the statements in the abstract following the reviewer’s comment. Regarding ECTV-MK, ‘We identified a putative strong inhibitor of the host inflammatory response in ECTV-MouKre, an isolate that did not cause local foot swelling and developed a moderate virulence.’ In the case of ECTV-H, ‘Most of the ECTVs, except ECTV-Hampstead, encode a truncated version of the P4c protein that impairs the recruitment of virions into the A-type inclusion bodies, and our data suggest that P4c may play a role in viral dissemination and transmission.’

Minor Comments:

Introduction

-For readers who are unfamiliar, I believe it would be good to briefly describe how virulence/pathology is greatly altered by mouse strain.    

Response: We added more details about mouse strain experiments in the Results (lines 225-228): ‘ECTV can infect all laboratory mouse strains but causes different severity depending on the mouse strain and the route of infection (59). C57BL/6 and AKR strains are resistant to severe disease by s. c. infection in the footpad, whereas A, BALB/cJ, DBA, and C3H are susceptible to severe disease by this route (60-63).’

Materials and Methods

-Use standard nomenclature for Balb/c mouse when describing source

Response: We have replaced Balb/c with BALB/cJ in the text.

-Provide more details for plaque assay in methods

Response: We added more details for plaque assay in section 2.1 Cells and viruses in Materials and Methods.

-I believe it would help to have another column in Table 1 briefly indicating source of each strain for clarity

Response: We have added this information in the new Table 1.

Results

-Line 232: Data not shown is discouraged in Viruses and should be included as supplemental.

Response: We now show the results of infection of DBA/2 mice with ECTV-M, ECTV-N, ECTV-H and ECTV-I in a new Supplementary Table (Table S2).

-Figure 1: Any statistical differences for weight loss (Please include)?  Please include that statistical analysis was at least done if so.

Response: No statistical analysis was done for weight loss. In cases were weight loss was clear, such as in ECTV-M1, loss was so dramatic as compared to control that we did not consider necessary to corroborate it by statistical analysis.

-Figure 1: It would be good to include some table or even a number on the swelling graph indicating how many mice lost feet since you mention it. 

Response: As data is shown as average, it is easier to clarify in the text which mice lost or did not lose the foot. All mice, except for one infected at 104 pfu, infected with ECTV-Ishibashi (at all doses) lost the feet. Only one mouse infected with106 pf ECTV-MH (see lines 258-260).

-Figure 2: Please indicate what log this is for virus titer.  It is also unclear what the authors mean by a single infection cycle.  Please clarify.

Response: We have added that the virus titer is expressed in Log10 in the Figure legend, and we have clarified that single infection cycle is one-step growth curve experiment by replacing them in the text (lines 134 and 294) and in the Figure legend.

-Figure 4: It is unclear what is being presented in 4A.

Response: Figure 4A shows a dot plot analysis which compares two sequences by organizing one sequence on the x-axis, and another on the y-axis, of a plot. When the nucleotide of both sequences match at the same location on the plot, a dot is drawn at the corresponding position. We added the description of the dot plot in the Figure legend.

-Line 462: Authors refer to popliteal ganglia?  Do you mean lymph node?  Please show data in Figure 6.

Response: We were referring to lymph node and we have corrected it in the text. We also indicated this in the Figure 6 legend.

-Figure 6: B and C labels are swapped.  Please include survival data in figure.

Response: We apologize for this mistake. We have corrected the labels in Figure 6, and also added the survival data in Figure 6C.

Reviewer 2 Report

Overall, the manuscript was very well researched information on historical ECTV and lineages. Comparison testing in vivo and sequencing provided strong evidence for the conclusions drawn. 

Please see a few specific suggestions/questions below. 

Line 65: Please clarify the year, or state '1950s'. 

Lines 119-120: Was the virus plaque purified from cell passage or in the host?

Line 124: Please define w/v.

Line 126: Was FBS heat-inactivated?

Line 127: Please define PFU. Suggest defining as MOI as well for consistency with rest of paper. 

Line 130: Suggest including a table or figure defining group numbers with N per group and control animals, days of infection, exposure, euthanasia, tissue collection, etc. for ease of understanding for the reader. 

Line 149: Define MOI. 

Line 150: Is this 1X or 10X PBS? 

Line 156: Please clarify and provide details to determine 'clearly infected'. 

Line 235: Is there any suggested explanation for death of the mouse if not from ECTV? Was this mouse included in analysis? 

Line 297: Please include percentages. 

Line 642: Please add accession number(s).

Overall comment: The sex of BALB/c mice can contribute to infectivity. Suggest including some discussion of this as only female mice were used. 

Author Response

Reviewer 2

Overall, the manuscript was very well researched information on historical ECTV and lineages. Comparison testing in vivo and sequencing provided strong evidence for the conclusions drawn. 

Please see a few specific suggestions/questions below. 

Line 65: Please clarify the year, or state '1950s'. see line 76.

Response: We have changed “fifties” for “1950s” as per reviewer’s suggestion (line 65).

Lines 119-120: Was the virus plaque purified from cell passage or in the host?

Response: ECTV-M and ECTV-N were plaque purified in cell culture, and this information is now included in the new Table 1.

Line 124: Please define w/v.

Response: we added the definition of the abbreviature in the text (weight/volume) (line 130).

Line 126: Was FBS heat-inactivated?

Response: Yes, and we have now indicated this in Material and Methods (lines 128-129). We removed the abbreviation for FBS.

Line 127: Please define PFU. Suggest defining as MOI as well for consistency with rest of paper. 

Response: we have defined pfu (line 135) and MOI (line 134) in the text, and removed the abbreviation of MOI.

Line 130: Suggest including a table or figure defining group numbers with N per group and control animals, days of infection, exposure, euthanasia, tissue collection, etc. for ease of understanding for the reader. 

Response: We have added the information about N of animals for Figure 1 and survival in new Supplementary Table (Table S1). The exposure, tissue collection for the transmission experiments (Figure 6) are depicted in Figure 6B.

Line 149: Define MOI. 

Response: We have removed the abbreviation MOI.

Line 150: Is this 1X or 10X PBS? 

Response: It was 1X and we added this to the text, see line 158.

Line 156: Please clarify and provide details to determine 'clearly infected'. 

Response: Clearly infected as harboring virions in the cell. We added this to the text in line 165. 

Line 235: Is there any suggested explanation for death of the mouse if not from ECTV? Was this mouse included in analysis? 

Response: We did not investigate the death of the mouse. We think it is better to include this mouse in the results, since we infected it with ECTV, and to explain that its death is likely to be unrelated to the infection since it occurred too early.

Line 297: Please include percentages. 

Response: We added the % for ECTV-H and ECTV-HE in the text. All the other isolates did not show any ATI embedded with virus, so the % would be zero (line 312).

Line 642: Please add accession number(s).

Response: We added the accession numbers for the isolates. Please note that ECTV-HE, ECTV-MH, ECTV-I, ECTV-M4, ECTV-M5 and ECTV-MK were submitted as a project and they have the same accession number in ENA (lines 667-670).

Overall comment: The sex of BALB/c mice can contribute to infectivity. Suggest including some discussion of this as only female mice were used. 

Response: We appreciate the need to address the possible influence of sex on susceptibility to infection. There are few reports addressing this issue, one for example showing a small increase (2-fold) of female mice to VACV intraperitoneal infection (Geurs et al. 2012. J. Autoimmun. 38:J245). To our knowledge, the sex of BALB/c mice does not have a major effect on the severity in the orthopoxvirus infection models we have used. We have not observed a major effect of mouse sex after intranasal infection with VACV or subcunaneous infection with ECTV in BALB/c mice (Deonarain et al. 2000. J. Virol. 74:3404; Desdin-Mico et al. 2020. Science 368:1371). More experiments with larger mouse numbers will be required to identify minor differences in some models of infection. We have included a sentence in the Discussion to mention this possibility (lines 566-568).

Reviewer 3 Report

Overall, this is a well-written and well-designed article. Many biologists based on the study the highly attenuated strains believe that ECTV is dangerous only for some laboratory mouse strains. Mavian et al. showed that ECTV strains demonstrate different pathogenesis and can differ significantly from each other. Mavian et al. have monitored the pathogenesis of nine ECTV strains and report the sequence of eight novel ECTV strains, including the first ECTV isolated from a field mouse.

Minor comments:
1. Line 82: ECTV has a narrow host range restricted to the mouse, its natural host…

Meyer et al. [Neubauer, H., Pfeffer, M., & Meyer, H. (1997). Specific detection of mousepox virus by polymerase chain reaction. Laboratory animals, 31(3), 201-205] write: «The seventh MPV strain (SF) had been isolated from a silver fox (Vulpes vulpes) bred on a farm in the Czech Republic (Mahnel et a1. 1993)». However, data on the pathogenicity of the ectromelia virus for fox is not available to me. The use of this information is at the discretion of the authors.

  1. Mavian et al. used the abbreviation "A-type inclusion bodies (ATIs)" throughout the text. It is more correct to write A-type inclusion (ATI) bodies (or proteins). At the same time, the text (line 158) uses the abbreviation ATI (without s).
  2. Line 615: “Therefore, ATIs might be a viral self-contention mechanism that ensures host survival and persistence in natural reservoirs, as for CPXV or the new world orthopoxviruses.” Apparently the authors mean the New World orthopoxviruses: raccoonpox, volepox and skunkpox viruses. I have no information about ATI bodies for skunkpox virus. In any case, a link to the source of information is required.

Author Response

Reviewer 3

Overall, this is a well-written and well-designed article. Many biologists based on the study the highly attenuated strains believe that ECTV is dangerous only for some laboratory mouse strains. Mavian et al. showed that ECTV strains demonstrate different pathogenesis and can differ significantly from each other. Mavian et al. have monitored the pathogenesis of nine ECTV strains and report the sequence of eight novel ECTV strains, including the first ECTV isolated from a field mouse.

Minor comments:
1. Line 82: ECTV has a narrow host range restricted to the mouse, its natural host…

Meyer et al. [Neubauer, H., Pfeffer, M., & Meyer, H. (1997). Specific detection of mousepox virus by polymerase chain reaction. Laboratory animals, 31(3), 201-205] write: «The seventh MPV strain (SF) had been isolated from a silver fox (Vulpes vulpes) bred on a farm in the (Mahnel et a1. 1993)». However, data on the pathogenicity of the ectromelia virus for fox is not available to me. The use of this information is at the discretion of the authors.

Response: We added this report (Neubauer et al. 1997, Ref. 38) in the Introduction (line 85).  

  1. Mavian et al. used the abbreviation "A-type inclusion bodies (ATIs)" throughout the text. It is more correct to write A-type inclusion (ATI) bodies (or proteins). At the same time, the text (line 158) uses the abbreviation ATI (without s).

Response: We have corrected the abbreviation throughout the text.

  1. Line 615: “Therefore, ATIs might be a viral self-contention mechanism that ensures host survival and persistence in natural reservoirs, as for CPXV or the new world orthopoxviruses.” Apparently the authors mean the New World orthopoxviruses: raccoonpox, volepox and skunkpox viruses. I have no information about ATI bodies for skunkpox virus. In any case, a link to the source of information is required.

Response: The reviewer is correct, and we removed the reference to CPXV and New World orthopoxviruses. We leave the sentence to discuss a possible role of ATI bodies in natural infections.

Reviewer 4 Report

Mavian et al. have greatly increased the number of complete ECTV genome sequences and have correlated them with phylogeny and virulence in mice. Overall, the data appear of good quality and the paper is clearly written for the most part. My chief caveat is in regard to the transmission experiment, which should be repeated as mentioned in comment 6 below. Otherwise only minor changes are needed.

Comments

  1. 2B. It looks as if the 0-time titer differs by 2 logs for some viruses.
  2. 3A. Describe the meaning of the white box in ECTV-HE and whether the vertical lines indicate single nucleotide differences.
  3. 5A. Specify in the legend whether there are any mutations that cause termination of the reading frame.
  4. 5B. It looks as if the ATIp ORF is the same length in all isolates
  5. I could only find one supplementary table that is labeled Table S2. “Absent” is spelled incorrectly. Also, difficult to match up text and Table. I could not find EVI003. It would help to add an additional column that indicates the function of the genes
  6. Fig. 6. B and C mislabeled. The transmission experiment should be repeated because only 2 infected mice were used for each time interval, and the difference was only found in one time period. The statistical analysis is for the number of mice that were infected whereas the number of mice that can transmit is also of importance.
  7. Line 520. “mice” should be changed to “a mouse”
  8. Lines 520-523. Does the date of isolation and date of outbreak suggest whether transmission went from laboratory to wild or wild to laboratory?

Author Response

Reviewer 4

Mavian et al. have greatly increased the number of complete ECTV genome sequences and have correlated them with phylogeny and virulence in mice. Overall, the data appear of good quality and the paper is clearly written for the most part. My chief caveat is in regard to the transmission experiment, which should be repeated as mentioned in comment 6 below. Otherwise only minor changes are needed.

Comments

1. 2B. It looks as if the 0-time titer differs by 2 logs for some viruses.

Response: We observed a one log difference at time 0 which may reflect the variability in the inoculum that was between 3.5-10 pfu/cell.

2. 3A. Describe the meaning of the white box in ECTV-HE and whether the vertical lines indicate single nucleotide differences.

Response: We added the information in the figure legend.

3. 5A. Specify in the legend whether there are any mutations that cause termination of the reading frame.

Response: The mutations that cause termination of the reading frame are represented by the light blue color and cause “partial deletion” of the protein. We have added this in the Figure legend.

4. 5B. It looks as if the ATIp ORF is the same length in all isolates

Response: The reviewer is correct. The ATI ORF is the same length in all the isolates, and therefore all isolates produce ATI bodies. However, full length P4c is only present in ECTV-H and ECTV-HE and therefore allows these viruses to be recruited into the ATI bodies. By contrast, the other isolates produce ATI bodies that remain empty.

5. I could only find one supplementary table that is labeled Table S2. “Absent” is spelled incorrectly. Also, difficult to match up text and Table. I could not find EVI003. It would help to add an additional column that indicates the function of the genes

Response: We apologize for the mistake. EVI003 is the homologue of EVM002 as it follows ECTV-N nomenclature. We have added more information on genes and function in the Supplementary Table (Table S3), and hope it will help the reader.

6. Fig. 6. B and C mislabeled. The transmission experiment should be repeated because only 2 infected mice were used for each time interval, and the difference was only found in one time period. The statistical analysis is for the number of mice that were infected whereas the number of mice that can transmit is also of importance.

Response: We have corrected the labels B and C. The transmission experiment was repeated for Group 2, and in Group 3 the 2 mice transferred to a cage with naïve mice were also able to transmit the virus. We believe that most of infected mice did show transmission, except in Group 1 where we found that the time of infection was too early to cause transmission to naïve mice. We believe the experiment shown demonstrates the ability of ECTV-H and ECTV-M to transmit at different times of infection. Also, our BSL3 facility is now dedicated to COVID-19 projects and it may take months before we have a chance to repeat this experiment.

7. Line 520. “mice” should be changed to “a mouse”

Response: We have corrected it in the text (line 544).

8. Lines 520-523. Does the date of isolation and date of outbreak suggest whether transmission went from laboratory to wild or wild to laboratory?

Response: Both scenarios are plausible, but based on solely the date of outbreak it is possible to draw conclusions on the origin and the ECTV-M1/-M4/-M5 outbreaks from laboratories were all predating ECTV-MK, the only isolate from wildlife.

Round 2

Reviewer 1 Report

The authors have addressed my major concern of mouse experiment reproducibility as well as other minor comments.  I believe this manuscript is acceptable for publication in Viruses.